# Asthma: The Use of Animal Models and Their Translational Utility

**DOI:** 10.3390/cells12071091

**Published:** 2023-04-05

**Authors:** Jane Seymour Woodrow, M. Katie Sheats, Bethanie Cooper, Rosemary Bayless

**Affiliations:** 1Department of Clinical Studies, New Bolton Center, College of Veterinary Medicine, University of Pennsylvania, Kennett Square, PA 19348, USA; 2Comparative Medicine Institute, College of Veterinary Medicine, North Carolina State University, Raleigh, NC 27606, USA

**Keywords:** asthma, animal model, inflammation

## Abstract

Asthma is characterized by chronic lower airway inflammation that results in airway remodeling, which can lead to a permanent decrease in lung function. The pathophysiology driving the development of asthma is complex and heterogenous. Animal models have been and continue to be essential for the discovery of molecular pathways driving the pathophysiology of asthma and novel therapeutic approaches. Animal models of asthma may be induced or naturally occurring. Species used to study asthma include mouse, rat, guinea pig, cat, dog, sheep, horse, and nonhuman primate. Some of the aspects to consider when evaluating any of these asthma models are cost, labor, reagent availability, regulatory burden, relevance to natural disease in humans, type of lower airway inflammation, biological samples available for testing, and ultimately whether the model can answer the research question(s). This review aims to discuss the animal models most available for asthma investigation, with an emphasis on describing the inciting antigen/allergen, inflammatory response induced, and its translation to human asthma.

## 1. Introduction

Asthma is a heterogeneous disease that is common in humans, affecting 1–18% of the population in various countries [1]. Asthma is characterized by chronic airway inflammation, airway remodeling, bronchial hyperreactivity, and partially reversible airflow obstruction [2,3,4,5]. The long-term goals of asthma management are aimed at controlling symptoms so that the individual can maintain normal activities, as well as avoid exacerbations, persistent airflow obstruction, and asthma-related death. Treatment primarily relies on antigen avoidance and medications such as corticosteroid and bronchodilator therapies. The treatment and management of asthma have become more individualized, especially as the disease itself is so heterogeneous. Understanding the pathophysiology of asthma is therefore critical to individualized medicine and the achievement of long-term management goals. Asthma in humans can be defined by endotype and/or phenotype. While there are multiple asthma phenotypes, the endotype is broadly divided into T2 high and non-T2 [6]. The T2-high asthma endotype involves allergy-mediated responses that involve eosinophils, interleukin (IL)-4, IL-5, IL-13, IgE, thymic stromal lymphopoietin (TSLP), leukotrienes, and prostaglandin D2 (PGD2) [7,8]. The T2-low or non-T2 asthma endotype is seen with airway neutrophilia or paucigranulocytic inflammation and is thought to be a mixture of Th1 and Th17 immune responses. Discrete divisions between these endotypes do not exist in every patient, hence the heterogenous disease process. With the greater availability of optimized animal models, the T2-high endotype has been more extensively researched, is better characterized, and has more biological therapy options including those that target IgE, eosinophils, IL-4, IL-5, and TSLP [9,10].

Animal models have been and continue to be essential for the discovery of molecular pathways driving the pathophysiology of asthma and for the discovery of novel therapeutic approaches. The use of cell lines can help to establish basic pathways in vitro, but these approaches cannot replicate intricate in vivo interactions. With the use of animal models, significant advances have been made toward the better understanding and treatment of the T2-high asthma endotype. The non-T2 asthma endotype, on the other hand, is less well characterized and has comparatively few treatments. Part of this lack of understanding is due to the fewer relevant animal models available for research. Recently, there have been several reviews about animal models of asthma [11,12,13,14,15]. This review article aims to summarize the available animal models used for asthma research and to describe their translational relevance to human asthma. Special emphasis is placed on naturally occurring animal models of asthma and how veterinarians, physicians, and researchers can be collaborative in our overall goal of better defining asthma pathophysiology, diagnosis, and treatment. This review article is not aimed at critically evaluating the specific protocols, especially those in mice, used to investigate asthma and is not a meta-analysis of data derived from animal models of asthma. The search strategy used for this review included using databases PubMed (1946 to present), CAB abstracts (1910 to present), and Google Scholar (1995 to present). Keywords included asthma, airway, inflammation, animal model, mouse, murine, rat, rabbit, sheep, ovine, guinea pig, dog, cat, horse, non-human primate, and human in various combinations. The primary literature and review articles were included.

## 2. Inducible Animal Models of Asthma

Most animal species, other than humans, cats, and horses, do not develop asthma naturally or spontaneously. Therefore, most animal species used to research and model asthma require human intervention to mimic the disease. The species most used in disease models is the mouse due to its short gestation, multiple inbred strains, ease of gene manipulation, and availability of reagents. A variety of models could be used depending on the research question of interest, including, mice, rats, guinea pigs, rabbits, ferrets, dogs, sheep, and non-human primates. Inducible models of asthma typically require abnormal sensitization routes and the use of adjuvants for an adequate immune response. Additionally, inducible models are not useful for longitudinal studies or to study aspects of asthma related to chronicity as most of the animal models undergo desensitization.

### 2.1. Mice

Mice are the most widely used animal model for the investigation of many diseases, including asthma. In 1994, a mouse model of allergic pulmonary inflammation was described using chicken ovalbumin (OVA) [16]. Benefits of mouse models include the availability of genetically engineered transgenic or gene-knockout strains, short gestation (20–30 days), and readily available species-specific reagents. Decades of research in mice have greatly enhanced our understanding of asthma and especially that of the T2-high asthma endotype [17,18,19]. OVA is inexpensive, has well-described MHCI and MHCII epitopes, and there are genetically engineered transgenic mice available for investigating OVA-specific responses. Other allergens, which can be argued are more natural stimulants of T2-high asthma development or exacerbation, are house dust mites (HDMs), fungal elements such as *Alternaria alternata*, pollen, and cockroach extracts [20,21,22,23,24,25]. Because OVA is not an important allergen in human asthma, these other allergens offer more clinical and pathophysiological relevance. Humans with asthma are noted to display exaggerated asthmatic episodes upon inhalation of HDMs [26]. In mouse models, chronic exposure to HDMs may replicate some aspects of chronicity and airway remodeling, as well as be useful for assessing treatments [27,28,29]. Additionally, the route of administration of the sensitizers is also variable in mice depending on the model, from intraperitoneal, subcutaneous, intra-tracheal, and aerosolized. Intraperitoneal and subcutaneous routes are least similar to the spontaneous development of asthma, which is due to chronic airway exposure to the antigen. Important viral pathogens that have been associated with asthma exacerbations in humans are influenza, rhinovirus, and respiratory syncytial virus, and mouse models have been useful for studying the impact of viral infections on asthma pathology [30,31,32,33,34,35]. 

The non-T2 asthma endotype and chronic asthma have been more difficult to study with the use of mouse models, as mice become tolerant to antigen exposure. The various mouse strains, sensitization, challenge methods, and type of asthma modeled (acute versus chronic) have been reviewed extensively, with a few specified here [29,36,37,38,39,40]. Many of the models available are acute, and the longest duration for a chronic model is approximately 12 weeks. When evaluating the response to the allergen challenge, many are T2-high asthma models, even when used chronically. This emphasizes the limited ability to investigate the pathophysiology of non-T2 asthma, especially in a chronic model, using a mouse model. Table 1 summarizes common mouse models of asthma. When evaluating the mouse strains used, most commonly BALB/c or C57BL/6 mice are used. BALB/c mice are IgE-high responders/Th2 immune responders, while C57BL/6 mice are skewed more to a Th1 immune response and are low-IgE producers [41,42,43,44,45]. Humanized mouse models can also be utilized, which allows for the investigation of various components of the human immune system [46,47,48]. Non-T2 asthma mouse models are limited, but neutrophilic airway inflammation has been able to be developed [38,49,50]. Most of the non-T2 asthma models rely on adoptive transfer of specifically differentiated Th cells. The inflammation though is short-lived, making longitudinal studies of either eosinophilic or neutrophilic airway inflammation impossible in the mouse. Additional important differences between mouse asthma models and naturally occurring human asthma include the transience of methacholine-induced AHR post-allergen exposure in mice, the desensitization that occurs with repeated allergen exposure in mice, and the variable need for IgE and mast cells in mice [14,37,40,51].

**Table 1 cells-12-01091-t001:** Mouse models of asthma, acute versus chronic.

Strain	Allergen	Sensitization	Exposure/Challenge	Pulmonary Inflammation	References
BALB/c	OVA	OVA (IP) on 7 alternate days	OVA aerosol for 8 consecutive days	Acute	[52,53]
BALB/c	OVA	OVA (IP) on 7 alternate days	OVA (IT) on day 42 for 3 days, each 3 days apart	Acute	[54]
BALB/c	OVA	OVA + AlOH_3_ (IP) on day 0 and OVA (IP) on day 10	OVA aerosol on days 17 and 24	Acute	[55]
BALB/c	OVA	OVA + AlOH_3_ (IP) on days 0 and 5	2 × OVA inhalations, each 4 h apart on day 17	Acute	[56]
BALB/c	OVA	OVA + alum (IP) on days 0 and 14	OVA (IN) on days 14, 25, 26, and 27	Acute	[57]
BALB/c	OVA	OVA + AlOH_3_ (IP) on days 0 and 14	OVA aerosol on days 28–30	Acute	[58,59]
BALB/c	OVA	OVA/alum (IP) on days 0 and 12	OVA aerosol on days 18–23	Acute	[60,61,62]
BALB/c	OVA	OVA + alum (IP) on days 0 and 14	OVA aerosol on days 28–30 and 72 days after last challenge	Acute	[63]
BALB/c	OVA + LPS	OVA-specific Th1 (IV)	Day 1 OVA (IN) daily for 4 days, last treatment day LPS (IN)	Acute	[49]
BALB/c	OVA	OVA (IT) on day −1, day 0 Th17 retro-orbital	Day 1–3 daily OVA (IT)	Acute	[50]
C57BL/6	HDM	Der p 1 + AlOH_3_ (IP) on day 0	HDM aerosol on day 14 for 7 consecutive days	Acute	[64]
C57BL/6	OVA + LPS	OVA + LPS (OP) on days 0 and 7	OVA (OP) on days 14–16	Acute	[65]
A/J	Bla g 2 and Der f 1	Ova + AlOH_3_ (IP) on days 0 and 7	Allergen oro-tracheal on day 14	Acute	[66]
C57BL/6	OVA	OVA-DCs (IT) on day 0	OVA aerosol on day 14–20	Acute	[67]
CB.17 SCID	Dpt	Human PBMCs (IP)	Dpt aerosols 1×/day for 4 consecutive days starting day 14	Acute	[46]
NOD/SCID	HDM	Human PBMCs (IP)	Dpt (IT) on days 1, 3, 7	Acute	[47]
BALB/c	OVA	Ova + AlOH_3_ (IP) on days 0 and 5	OVA aerosol 3 days/week, starting day 17, for 6 weeks	Chronic	[56]
BALB/c	OVA	OVA + alum (IP) on days 7 and 21	OVA exposure 3 days/week, for up to 8 weeks	Chronic	[68]
BALB/c	OVA	OVA + alum (IP) on days 0 and 14	OVA (IN) on days 14, 27, 28, 47, 61, and 73–75	Chronic	[69]
BALB/c	OVA	OVA + aluminum potassium sulphate (IP) on days 1 and 11	OVA (IN) on days 11, 19, 20, 33, 34, 47, 48, 61, 62, 75, 76, 89, 90	Chronic	[70]
BALB/c	OVA	OVA + alum (IP) on days 0 and 12	OVA aerosol on days 18–23 and then 3 days/week for up to 8 weeks starting on day 26	Chronic	[62]
BALB/c	OVA	OVA + alum (SC) on days 0, 7, 14, 21	OVA (IN) on days 27, 29, 31, and then 2×/week for 3 months	Chronic	[71]
BALB/c	HDM	NA	HDM (IN) 5 days/week for up to 7 weeks	Chronic	[27]
BALB/c	HDM	NA	HDM (IN) 5 days/week for up to 5 weeks	Chronic	[28]

IP = intraperitoneal, IN = intranasal, IT = intratracheal, SC = subcutaneous, IV = intravenous, OP = oropharyngeal aspiration, NA = not applicable.

In addition to the previously mentioned benefits of reagent and assay availability and the variety of genetically engineered transgenic mice, terminal mouse asthma studies also allow for easy tissue collection. Lung function testing is also available in the mouse and recent advancements have made it possible to assess lung function longitudinally, rather than terminally [72,73,74,75]. Like other inducible models, the pathophysiologic relevance of murine asthma models to naturally occurring human asthma has been questioned. Murine asthma models are also limited in their ability to model the chronicity of human asthma. Anatomically, mice have monopodial branching of the bronchi while humans have dichotomous branching; mice have zero to one respiratory bronchioles while humans have several generations; the airway cartilage is limited to the trachea in mice while it extends from the trachea to the distal bronchioles in humans [76]. The airway inflammation of mice in asthma models may also more similarly reflect allergic alveolitis than asthma, and the roles of eosinophils and mast cells vary between mice and men. Mouse eosinophils, in OVA-induced airway inflammation, do not have significant degranulation [77]. Additionally, both histamine and serotonin play a role in AHR in mice and are released from mast cells, while serotonin’s role in human AHR is unclear [78]. While limitations and differences exist, mouse models of asthma are likely to remain the most used animal model for investigating the pathophysiology and molecular pathways of asthma. Complimentary studies in naturally occurring animal models of asthma could be used to further validate and support the relevance of findings in inducible models such as mice.

### 2.2. Guinea Pigs

Asthma is not a naturally occurring disease in guinea pigs, but the immediate hypersensitivity reaction of the lungs has been appreciated for some time in guinea pigs [79]. The use of OVA as a sensitizer for the study of T2-high asthma is common in inducible animal models, such as mice and rats. Guinea pigs can also be sensitized to ovalbumin, or other stimulants, to induce IgE-mediated airway mechanisms that replicate a response similar to the asthma phenotype of humans that involves eosinophilia and increased airway responsiveness [80,81,82,83,84,85]. OVA can be delivered via various routes for sensitization, including peritoneal, subcutaneous, and aerosolized. Low-dose OVA, 10 µg, has been shown to induce early asthmatic responses (EARs) with the production of IgE and IgG_1_ [86,87]. A larger dose of OVA, 100 µg, can induce both EAR and late asthmatic responses (LARs) [86,87,88,89]. Aerosol sensitization, more physiologically like naturally occurring asthma, with low-dose OVA induces EAR and LAR with IgE and IgG_1_ production and high-dose OVA induces EAR, airway eosinophilia, and airway hypersensitivity [90,91,92]. The use of the allergen house dust mite (HDM) has also been shown to induce airway hyperresponsiveness, eosinophilia, and the recruitment of mast cells, which, in repeated intranasal HDM exposure, additionally caused airway wall hyperplasia [93].

The acute, allergic hypersensitivity reactions in the airway of asthmatic humans, such as airway smooth muscle contraction, eosinophil infiltrate, airway hyperresponsiveness, and mucus production, are mediated in part by activation of the histamine H1 receptor and leukotriene cysteinyl leukotriene (cysLT)-1 receptor [94,95,96,97]. These receptor activations are also appreciated in guinea pigs [98,99,100]. Given the similarities, guinea pigs have been used to develop drugs used in the treatment of asthma. Leukotriene receptor antagonists, such as Montelukast, were developed by the use of guinea pig models [101,102,103,104,105,106,107,108]. A phosphodiesterase (PDE3/4) inhibitor, ensifentrine, aimed at relaxing airway smooth muscle was also developed in guinea pigs and its function was confirmed in humans [109,110,111].

While there are similarities and proven benefits of guinea pigs as a research model for asthma, there are differences worth acknowledging. Guinea pigs have an elongated soft palate and, as a result, are obligate nasal breathers [112]. They have seven lung lobes, three right and two left, and two accessory lobes. They have a dichotomous branching of the bronchial tree, as do humans, although guinea pig and human branching differ slightly, with guinea pigs having fewer bifurcations. The parenchyma of the lung lacks connective tissue and is therefore more fragile than human lung. The alveoli contain many macrophages, but phagocytosis appears to be primarily via neutrophils that move into the alveoli. Chronicity is not a feature of induced asthma in guinea pigs, as they become tolerant to the allergens used and do not display nonspecific hypersensitivity [113,114]. Additionally, bronchoconstriction in the guinea pig asthma models appears to be mediated by primarily by histamine, which may have limited translational relevance to human asthma patients in which antihistamines have limited efficacy [115]. Logistic limitations for guinea pigs as an animal model are a longer gestation (60–75 days) compared to mice and less reagent availability. There has been an effort to develop more guinea pig assays, identify translatable markers, and review the availability of monoclonal antibodies to make research more viable in this species [116,117,118,119,120,121]. 

### 2.3. Rabbits

Rabbits were among the first animal models for asthma research. Rabbits are phylogenetically more similar to primates than other rodents. Similarities between induced asthma in rabbits and humans include bronchoconstriction, airway obstruction, and airway hyperresponsiveness [122,123]. Rabbit models of asthma require antigen sensitization within 24 h of birth for the development of late-phase airway responses when later challenged; however, this is not necessary if only investigating early-phase airway responses [122,124,125,126,127,128]. IgE is the primary anaphylactic antibody response for rabbits. Additionally, rabbits have both heterophils, which are analogous to human neutrophils, and eosinophils; special staining is required to differentiate these cells from one another [129]. Lung function testing can be performed for more objective measurements, and histamine or methacholine can be used for evaluating airway responsiveness [130,131,132]. Asthma treatments such as corticosteroids, PDE4 inhibitors, bronchodilators, and others of various routes of delivery have variable effectiveness in rabbit models [133]. A factor that may influence the deposition/distribution of air and particles in the lung is the fact that rabbits have monopodial branching, different from humans. Rabbits also lack submucosal glands and goblet cells appear to be less numerous than those in humans [134]. Limitations to their use is reagent availability, the low number of transgenic lines, and increased cost. 

### 2.4. Sheep

A model of maternal allergic asthma in sheep was recently developed as a method to investigate the impacts of the intrauterine environment on asthma in both mothers and offspring [135,136,137,138]. Immune consequences of the allergic airway inflammation of humans are replicated in sheep, including IgE-related responses, the recruitment of eosinophils and lymphocytes, locally activated mast cells, and mucus production [139,140,141]. Both early- and late-phase allergic airway responses have been noted in sheep models [142,143]. Allergic airway inflammatory responses in sheep have been utilized for evaluating novel airway treatments, such as Montelukast and tryptase inhibitors [108,144]. However, not all novel therapeutics extrapolate to humans, for example, inhibitors of platelet-activating factors are effective in sheep, but not humans with asthma [145]. Multiple studies have also utilized sheep for the investigation of Th2-driven asthma using HDM and to study the impact of chronic lower airway inflammation on lung function and airway remodeling [140,143,146,147,148]. Lung function testing can be performed on sheep for additional objective data analysis [149]. Sheep lobes are also well separated by tissue septa, making it easier to apply different treatments to localized areas [150]. Airway branching in sheep, similar to other large mammals and humans, is dichotomous [150,151]. Additionally, large volumes of blood can be collected from sheep, compared to other inducible models. Compared to rodent models though, the cost of sheep as a translation asthma model is relatively high.

### 2.5. Rats

Rat asthma models most commonly use OVA for sensitization and the strain most utilized is the Brown Norway rat [152,153,154,155]. There are also models described using HDM as an antigen [156]. The HDM model has been used more recently for evaluation of treatments in the area of angiogenesis [157,158]. OVA, in combination with LPS, has been used to model eosinophilic, neutrophilic, and lymphocytic airway inflammation and lung remodeling [159]. Rats produce IgE; display hyperreactivity to methacholine, acetylcholine, and serotonin; accumulate neutrophils, lymphocytes, and eosinophils in BALF; show elevations in Th2 cytokines similar to allergic asthma in humans [160,161,162,163]. Rats have a weak bronchoconstriction response and require high levels of antigen/agonist exposure to elicit this response. Regarding treatments, beta2-receptor agonists (such as salbutamol) and steroids (dexamethasone and budesonide) are effective in rat models [164,165,166]. 

### 2.6. Dogs

Dogs develop allergic diseases, but atopic dermatitis is much more common than asthma; therefore, it is not feasible to use dogs as a naturally occurring animal model of asthma. However, an inducible model of asthma in dogs has been reported. Several studies have been performed in dogs being naturally or actively sensitized with *Ascaris suum* larvae investigating the role of the vagus and phrenic nerve, COX responses, and airway responses to molecules such as histamine, prostaglandins, and acetylcholine [167,168,169,170,171,172,173,174,175,176]. Ragweed-sensitized beagles are another popular model used to study mechanisms of asthma [177,178,179,180,181]. Interestingly, in dogs, neutrophilic airway inflammation is more common than eosinophilic inflammation following allergen challenge [182,183,184,185,186]. Additionally, dogs have the unique ability to develop prolonged airway hypersensitivity, up to 5 months, to *A. suum* [187]. Advantages of dogs as an animal model include its size and ability to collect large sample volumes, but reagents specific for dogs are limited compared to reagents for mice and humans.

### 2.7. Nonhuman Primates

Models of asthma in nonhuman primates have been utilized, primarily in rhesus monkeys and macaques [188,189,190,191,192,193,194,195]. Sensitizers most commonly used have been Ascaris extract, HDMs, and birch pollen allergens, all of which induce a Th2 immune response. Benefits of using nonhuman primates include genetic and physiological similarities to humans, including the ability to be upright and bipedal. Pulmonary lung function testing and bronchoscopy can also be performed [196]. In the case that nonhuman primates are euthanized for other reasons, precision-cut lung slices could be used for ex vivo modeling to investigate bronchoconstriction, including the response to therapies [197]. Access to reagents is easier in nonhuman primates compared to other large animal models. Disadvantages of using nonhuman primate models include high cost and ethical concerns.

## 3. Naturally Occurring Animal Models of Asthma

Animals that develop asthma spontaneously allow for the investigation of asthma development and pathophysiology in a ‘real-world’ scenario. Longitudinal studies, chronic asthma changes, and the heterogeneity of asthma can be investigated in these models. Translational clinical studies could also be pursued through collaborations with veterinarians. Both cats and horses develop asthma naturally. Cats develop T2-high asthma, while horses have T2-high and non-T2 asthma subtypes. 

### 3.1. Cats

Cats are one of the species that naturally develop asthma. Characteristics of feline asthma are very similar to humans, including features such as eosinophilic airway inflammation, bronchoconstriction, and airway remodeling [198,199,200,201]. While the exact pathophysiology of feline asthma is not well understood, it is believed to be an allergic-type etiology [202,203,204,205,206,207,208]. In support of allergy-mediated etiology, allergen-specific immunotherapy and allergen avoidance can help control clinical signs [203,206,209]. There can be an overlap to other diseases such as bronchitis and the limitation of not being able to perform lung function testing to evaluate for reversible airway obstruction after inhalation of a bronchodilator potentially leads to the confusion of asthma and bronchitis [210]. While clinical studies can be performed in naturally occurring cases of asthma in cats, there are also models that use specific antigen exposures such as with HDM and Bermuda grass [205]. A type 2 immune response profile has been identified in peripheral blood mononuclear cells and BALF cells in cats with asthma. An experimental induction of asthma in cats has also been used to evaluate microbiome changes of the airway, both in acute and chronic stages [211].

In addition to being a naturally occurring model, feline asthma has additional translational relevance because cats live in the same environment as humans and may be responding to similar airway antigens. Both humans and cats have alveolarized respiratory bronchioles, but these bronchioles continue for several generations in humans, while cats only have one generation [212,213]. Well-developed submucosal glands and ciliated epithelial cells of the airway appear to be comparable between humans and cats; therefore, they may have similar mucociliary clearance. Histologic changes seen in human asthma, such as airway wall inflammation, epithelial cell hyperplasia and desquamation, hypertrophy and hyperplasia of airway smooth muscle, and hypertrophy of submucosal glands, are appreciated in cats, except for a thickened basement membrane [205,214,215]. Similarities also exist in the non-adrenergic, non-cholinergic inhibitory nervous system control over airway diameter [216,217,218]. Cats display an airway hyperresponsiveness to methacholine and reversibility to bronchoconstriction with beta agonists, similar to humans [201]. Whole-body plethysmography can be performed in cats [219]. While a similar type 2 immune response and associated cell types, cytokines and chemokines, are seen in humans and cats with asthma, differences in histamine response exist, with cats displaying both dilation and constriction. Additionally, cysLT do not play an important role in cats as it does in human asthma [220,221,222,223].

### 3.2. Horses

Horses are an additional species that develop asthma naturally. Currently, equine asthma (EA) includes two categories: mild to moderate equine asthma (mEA) and severe equine asthma (sEA). Unlike human asthma, the term ‘severe’ for EA is used to describe a greater degree of lower airway inflammation and severity of clinical signs but does not necessarily mean the asthma is difficult to manage or treat. Equine asthma is a chronic inflammatory lung disease with characteristics similar to human asthma, including enhanced bronchial reactivity; chronic, partially reversible airflow obstruction; pulmonary remodeling; lower airway inflammation [224,225,226]. Equine asthma prevalence is not clearly defined, but some studies suggest that up to 80% of horses may have mEA and up to 17% may have sEA [227,228,229,230,231]. A 2016 consensus statement by the American College of Veterinary Medicine defined two categories of equine asthma (EA), mEA and sEA, and the diagnosis of each [224]. The diagnosis of asthma is based on history, clinical signs, physical examination, bronchoalveolar lavage fluid (BALF) cytology, and lung function tests, depending on availability. Clinical signs supportive of sEA include cough; increased respiratory rate and effort at rest; poor performance/exercise intolerance; increased tracheal mucus, crackles, and/or wheezes on lung auscultation; serous, mucoid, or mucopurulent nasal discharge. By contrast, clinical signs of mEA are primarily linked to athletic performance and can include cough, decreased performance, or prolonged respiratory recovery following exercise. Importantly, horses with mEA do not have respiratory abnormalities at rest. BALF is routinely collected for EA diagnosis. For mEA, consistent cytologic findings include mild to moderate increases in % neutrophils (5–20%), mast cells (>2%), eosinophils (>1%), or a combination of cell types. For sEA, cytology shows a marked increase in % neutrophils (>20–25%). Cultures and complete blood count are often used to rule-out pneumonia. The term paucigranulocytic inflammation is not widely used in veterinary medicine, but anecdotal evidence supports the relevance of this term for a subset of severely asthmatic horses. 

Asthma immune responses in both horses and humans have been broadly characterized by a Type 2 immune response, termed Type 2 high; or a mixed immune response, termed Type 2 low or non-type 2 [8,232]. The cytokine profiles of mEA and sEA, either at the protein or mRNA level, indicate a Th1, Th2, Th17, or mixed immune response, depending on the study [233,234,235,236,237,238,239,240,241,242,243,244,245]. Generally, samples from horses with mEA are most often a T2-high endotype, and samples from horses with sEA indicate a non-T2 endotype. Benefits of using the horse as a translational asthma model have been suggested by several review articles [246,247,248,249]. While the pathophysiology of both human and equine asthma is still being defined, they appear to have similarities regarding airway inflammation, remodeling, responses, clinical signs, and response to therapy. Horses display both acute and long-term airway responses. Because they are large animals, it is relatively easy to obtain large volumes of blood and airway lavage (tracheal and bronchial). In research settings, endobronchial brushings and/or biopsy samples, lung function testing, and even lung biopsies are frequently collected [250,251,252]. Due to the prevalence of EA and the importance and popularity of horses as performance athletes and companion animals, there is a need in veterinary medicine for improved treatments and diagnostics for EA. As a result, lung function testing, which has been technically difficult to perform in an ambulatory setting, has advanced to include portable testing options [253,254,255,256,257,258]. 

The primary trigger for EA is organic dust. Some asthmatic horses are sensitive to organic dust in barns and/or hay and experience worsening symptoms when housed indoors. Other asthmatic horses are sensitive to the organic dusts encountered in pasture environments during hot and humid conditions. This type of asthma is referred to as Equine Pasture Asthma (EPA) and is most commonly diagnosed in subtropical climates such as the southeastern United States. For either type of asthma, antigen avoidance is the most effective treatment for decreasing lower airway inflammation and symptoms. Therefore, horses who worsen indoors experience symptom relief when kept out of the barn environment and are fed soaked or steamed hay or a hay-alternative diet [259]; horses who worsen on a summer pasture experience symptom relief when moved indoors to a temperature-regulated environment [232]. In addition to environmental management, and as a necessary treatment during acute asthma exacerbation, horses with asthma are routinely treated with inhaled or systemic corticosteroids and bronchodilators [238,243,260,261,262,263]. There is also a demonstrated benefit to daily omega-3 supplementation in horses with sEA [264].

Like asthma in humans, asthma in horses is heterogenous with different clinical presentations (i.e., phenotypes) and immunopathogeneses (i.e., endotypes) [265]. Studies in horses with mEA provide variable evidence for the roles of mast cells, eosinophils and neutrophils, and Th2-high and Th2-low cytokines, suggesting that horses with mEA could be used as a translational model for several different human asthma endotypes. By contrast, studies in horses with sEA primarily show evidence for a mixed cytokine response (Th1/Th2/Th17) and consistent airway neutrophilia, suggesting translational relevance of sEA for non-allergic and/or neutrophilic asthma in humans [249,265]. Recent studies using differential gene expression analysis show striking similarities between the transcriptomic profiles of sEA and severe neutrophilic asthma in humans [266,267]. One of these studies also showed a significant upregulation of miR-142-3p and miR-223 in lung tissue samples from asthmatic horses vs. controls. These two miRNAs have also been shown to promote airway inflammation and obstruction in severe neutrophilic asthma in humans [266]. Historically, neutrophilic asthma in humans has been less well researched and understood, partially due to a lack of relevant animal models. This is despite neutrophilic asthma in humans being associated with greater morbidity and mortality [268,269]. While mouse models of neutrophilic asthma are now available [270], and have been used to identify novel biomarkers and therapeutic approaches [271,272], horses are another potentially useful translational model for this important asthma endotype. Like asthma in humans, sEA is a chronic disease that leads to structural changes in the lung that mirror changes in humans with asthma [225,273]. With environmental management, asthmatic horses can be maintained in remission and exacerbation can be induced when needed by feeding dusty/moldy hay or nebulizing hay dust extract (HDE). Clinical signs, blood samples, and airway samples including tracheal and bronchial lavage, brushings, biopsies, and pulmonary function can all be used for investigations of novel molecular pathways, identifications of new therapeutic targets, or when conducting short- or long-term clinical trials. For clinicians and researchers lacking equine expertise or facilities, it is possible to collaborate with veterinary clinician scientists who are already conducting translational and comparative research using EA as a model. Samples can also be requested from an Equine Respiratory Tissue Biobank (http://asthmeequin.com/en/facilities/, accessed on 22 September 2022). Equine asthma is an important spontaneous animal model of asthma with demonstrated value for future translational research.

## 4. Approach to Asthma Research

Using inducible animal models of asthma allows for large-scale research and increases the statistical power of the results. Mice are commonly the first in vivo model utilized due to gestation length, litter size, homogenous genetics, availability of transgenic mice, ease of terminal studies, cost, and access to species-specific reagents. While defining the pathophysiology of T2-high asthma and the development of novel therapeutics have greatly benefited from mouse models, non-T2 asthma research has been left behind. To define molecular pathways, the combination of in vitro, ex vivo, and mouse models are best utilized. The transgenic mice available for use enable specific aspects of a cellular pathway to be investigated. A step-up approach is likely best used when defining the pathophysiology of asthma and developing novel therapeutics. Findings in mice can be validated in other mouse models of asthma, such as other outbred strains or humanized mouse models, and once validated, the findings can be further investigated in more anatomically similar models. Finally, the hypothesized pathophysiology of asthma and novel therapeutics can be evaluated in naturally occurring cases of asthma, such as in cats and horses. One Health and fostering collaboration lend the opportunity to expand our research knowledge and ability to advance our understanding of asthma etiology, phenotypes, endotypes, diagnosis, and treatment. Both cat and horse owners are looking for more individualized medicine to best treat and manage their animal’s disease. Clinical research studies in veterinary species are common and veterinary patients are a group of animals with naturally developing disease in which physicians and basic researchers could collaborate. Longitudinal studies and features of asthma related to chronicity can be investigated in these naturally occurring models. Advanced research techniques are utilized in clinical trials in veterinary species and further collaboration can help establish similarities and differences in cats, horses, and humans with asthma. Additionally, horses offer a unique aspect for investigation as many horses are athletes, with athletic abilities spanning from local shows to Olympic/International events.

## 5. Conclusions

Asthma is a lifelong disease and requires constant medical management. Asthma puts a large economic burden on the individual and the country. In the United States alone in 2013, the estimated total cost of asthma, including costs due to missed work and mortality, was USD 81.9 billion [274]. A better understanding of the pathophysiology of asthma in humans by utilizing animal models of asthma is aimed at improving diagnostics and treatments. Depending on the research question and budget, a certain animal model of asthma may be more applicable for use. Naturally occurring animal models of asthma have the benefit of investigating the aspect of chronicity, collecting longitudinal data, and evaluating novel treatments. As summarized in Table 2, there are various pros and cons to each animal model of asthma and, ultimately, it is up to the researcher to determine which model is most appropriate to answer the research question.

**Table 2 cells-12-01091-t002:** Animal Models of Asthma.

Animal	Stimuli	Predominant Cell Type, Immunopathology	Characteristics Shared with Human Asthma	Strengths	Limitations
**Inducible Models**					
**Mouse** [17,18,19,37,275]	OVA +/− LPS, HDM, Cockroach, *Alternaria alternata* antigen, pollen	Eosinophils, Neutrophils Th2-high and Th2-low asthma models are available, depending on sensitization/challenge conditions	Bronchoconstriction Airway hyperresponsiveness Goblet cell hyperplasia/↑ Mucus production Airway smooth muscle hypertrophy Subepithelial fibrosis	Low cost Transgenic or gene-knockout strains Short gestation length Wide availability of mouse-specific reagents and assays	Lack of ability to model chronicity due to tolerance Limited/indirect measures of pulmonary function, generally requires anesthesia
**Guinea Pig** [85,276]	OVA, HDM	Eosinophils, Neutrophils Th2-high asthma, IgE-mediated	Bronchoconstriction, Airway hyperresponsiveness, Goblet cell hyperplasia/↑ Mucus production Subepithelial fibrosis	Low cost	Neutrophils are the primary phagocytes within alveoli (vs. alveolar macrophages) Lack of ability to model chronicity due to tolerance
**Rabbit** [122,123,277,278,279]	OVA, *Alternaria tenuis* antigen	Eosinophils Th2-high asthma, IgE-mediated	Bronchoconstriction, Airway hyperresponsiveness	Low cost	Sensitization can occur within 24 h of birth to optimize early and late airway responses Rabbits have heterophils instead of neutrophils
**Sheep** [140,143,146,147,280,281,282,283,284]	HDM, *Ascaris suum* antigen	Eosinophils Th2-high asthma, IgE-mediated	Bronchoconstriction Airway hyperresponsiveness ↑ airway collagen deposition ↑ bronchial smooth muscle thickness Goblet cell hyperplasia/↑ Mucus production	Similar placental physiology to humans Body size and anatomy allow pulmonary function assessment in conscious animals and large BAL fluid volume/cell number Lifespan is amenable to modeling chronic asthma	Cost Limited species-specific assays and antibodies
**Rat** [152,153,154]	OVA, HDM	Eosinophils Th2-high asthma, IgE-mediated	Bronchoconstriction, Airway hyperresponsiveness, Goblet cell hyperplasia/↑ Mucus production	Low cost Transgenic or gene-knockout strains Short gestation length Availability of rat-specific reagents and assays	Weak bronchoconstriction Require high levels of antigen exposure Limited/indirect measures of pulmonary function, generally requires anesthesia
**Dog** [285]	*Ascaris suum* antigen, Ragweed antigen, Ozone	Neutrophils, Eosinophils IgE-mediated	Airway hyperresponsiveness	Relative low cost among other large animal models	Limited species-specific assays and antibodies Public concerns with laboratory testing on companion animals
**Nonhuman Primates** [286,287,288]	Ozone, *Ascaris suum* antigen, HDM, Pollen, Tobacco smoke	Eosinophil Th2-high asthma, IgE-mediated	Bronchoconstriction Airway hyperresponsiveness ↑ Mucus production Subepithelial fibrosis	Upright, bipedal stance mirrors human posture to a greater extent than other quadruped species Greater overlap in airway transcriptome between rhesus macaques and humans Similarities in immune system compared to humans, many anti-human antibodies and reagents are cross-reactive with monkey antigens	Cost, Ethics of research involving nonhuman primates
**Naturally Occurring Models**					
**Cat** [205]	Dust/dust mites, Smoke (tobacco or fireplace), Pollen, Household chemicals	Eosinophils Th2-high asthma, IgE-mediated	Bronchoconstriction Airway remodeling	Indoor client-owned cats have similar environmental exposures to their owners Feline patients (no per diem fee) represent a recruitable population for testing novel therapeutics	Limited species-specific assays and antibodies
**Horse** [289,290]	Organic dust, Lipopolysaccharide, Fungal spores (e.g., *Aspergillus* sp.), Pollen	Neutrophils, Mast cells, Eosinophils, Mixed Th2-low and Th2-high asthma	Bronchoconstriction Airway hyperresponsiveness Airway remodeling Mucus production Bronchial angiogenesis	Spontaneous disease with similar triggers Pulmonary function testing can be performed at rest and during exercise Repeated collection of airway samples (large volume/cell recovery) and flexible bronchoscopy and long lifespan enable extended longitudinal studies (years ) Equine patients (no per diem fee) represent a recruitable population for testing novel therapeutics	Cost Limited species-specific assays and antibodies

## Data Availability

Not applicable.

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
