# Peer review of "Asthma: The Use of Animal Models and Their Translational Utility"

_cells, 2023, doi:10.3390/cells12071091_

Round 1
Reviewer 1 Report
1. 1. There are other relatively recent reviews on this subject uncited by the authors. The authors should cite these and then distinguish what they think their new review adds. (e.g., see doi 10.2147/JAA.S121092 or doi: 10.1002/jcb.24881)
2. 2. The paper needs more careful copyediting to week out misspellings: line 159 ‘scewed’ line 172 ‘anatomicall, anatomicall(y)’ ‘bonchi’
3. 3. I appreciate that the authors do not wish to advertise, but they should at least name some of the example drugs to which they refer. For example: lines 86-88: “Leukotriene receptor antagonists…such as monteleukast.”
4. 4. Mouse models: line 175 “Mouse models are likely to remain…” Yes. But the authors must reveal, unflinchingly, why that is so: study sections that approve grant funding are focused on molecular mechanisms, and it is cheaper to do these studies in mice for all the reasons the authors noted. Continuing to choose this species is reasonable for preliminary studies, but failure to replicate the initial observations from congenic to outbred strains and then onward to more anatomically relevant models is a major hurdle to developing effective therapies. Authors might wish to review an older set of point-counterpoint editorials re: the pros and cons of using mouse models in order to enrich their discussion of the limitations of the utility of this species. (see for example, E. Gelfand versus C. Persson DOI: 10.1164/rccm.2204001).
5. 5. After reviewing some of the more recent reviews on this subject, the authors may wish to propose a more systematic approach to using successive/complementary model systems to advance the field. Such a discussion would be an advance over some of the recently published reviews, and should be a natural exploitation of the veterinary expertise.
Author Response
Thank you for taking the time to comment and give suggestions for improvement of the manuscript.
- The additional reviews have been cited in Line 54. This review article aims to put special emphasis on the use of naturally occurring models of asthma, such as cats and horses, and promote collaboration with basic researchers and physician scientists (introduction, Line 56-58 and approach to asthma research, Line 525-554). One Health approach to asthma is likely to enhance our research findings.
- Misspellings have been identified and fixed throughout the manuscript.
- Specific drug names have been added, Lines 169, 170, 238, 259, 260.
- and 5. An 'approach to asthma research' section (starting Line 525) and discussion of a step-up approach was added in order to address these two very important points. Translational studies using veterinary species would be very welcomed in the veterinary community and an aspect that is likely not harnessed as well as it could be. Table 1 (newly made, Line 571) helps to emphasize the unnatural inducement of asthma and the lacking of chronic models in the mouse. Chronic lower airway inflammation is a part of defining asthma and really requires investigation using naturally occurring models of asthma. Additionally, Table 2 (Line 579) enables the reader to put the pros and cons together in a more simplified manner in order to compare animal models. Both of these Tables can be used as resources when deciding what animal model of asthma to utilize.
Reviewer 2 Report
This is a comprehensive review of experimental asthma models in different animal species.
Since the mouse is the major species for translational and fundamental research, I recommend to put the mouse in the first line addressing the use of different genetic backgrounds and expand list of allergen. Please also show protocols of acute, chronic and exacerbation models in the mouse in a Figure, which will interest more readers. The Table of the models should be condensed to be readable.
Author Response
Thank you for taking the time to comment and give suggestions for improvement of the manuscript.
Discussion of the mouse as a model of asthma was moved up in both the main text (Line 78) and in Table 2, animal models of asthma (Line 579).
Table 1, mouse models of asthma (Line 571), was added and further discussed in the main text (Line 109 - 118). Table 1 aims to give specific inducible protocols of asthma in the mouse, including genetics, allergens, and type of inflammation achieved (acute v. chronic and eosinophilic v. neutrophilic). Table 1 adds a lot to the review and can be a resource to use in choosing specific mouse models.
Table 2 was simplified, although still does not fit on one page. The Table could be split in to two separate Tables, inducible v. natural, but the inducible Table is likely to still be more than one page; therefore, it was left as one Table. The information still included in the Table are details the authors believe important to include.
Round 2
Reviewer 1 Report
The authors have made good faith efforts to address the previous criticisms. The further discussion about the exploitation of differing species' pathophysiology is an important improvement.